# The Mechanism of Manipulating Chirality and Chiral Sensing Based on Chiral Plexcitons in a Strong-Coupling Regime

**DOI:** 10.3390/nano14080705

**Published:** 2024-04-18

**Authors:** Xiongyu Liang, Kun Liang, Xuyan Deng, Chengmao He, Peng Zhou, Junqiang Li, Jianyu Qin, Lei Jin, Li Yu

**Affiliations:** State Key Laboratory of Information Photonics and Optical Communications, School of Science, Beijing University of Posts and Telecommunications, Beijing 100876, China; xiongyu_liang@bupt.edu.cn (X.L.); kunliang@bupt.edu.cn (K.L.); dengxy@bupt.edu.cn (X.D.); hcm990616@bupt.edu.cn (C.H.); simon_zhou@bupt.edu.cn (P.Z.); lijunqiang@bupt.edu.cn (J.L.); jyqin@bupt.edu.cn (J.Q.); leijin@bupt.edu.cn (L.J.)

**Keywords:** chiral plexction, strong coupling, circular dichroism, chiral sensing

## Abstract

Manipulating plasmonic chirality has shown promising applications in nanophotonics, stereochemistry, chirality sensing, and biomedicine. However, to reconfigure plasmonic chirality, the strategy of constructing chiral plasmonic systems with a tunable morphology is cumbersome and complicated to apply for integrated devices. Here, we present a simple and effective method that can also manipulate chirality and control chiral light–matter interactions only via strong coupling between chiral plasmonic nanoparticles and excitons. This paper presents a chiral plexcitonic system consisting of L-shaped nanorod dimers and achiral molecule excitons. The circular dichroism (CD) spectra in our strong-coupling system can be calculated by finite element method simulations. We found that the formation of the chiral plexcitons can significantly modulate the CD spectra, including the appearance of new hybridized peaks, double Rabi splitting, and bisignate anti-crossing behaviors. This phenomenon can be explained by our extended coupled-mode theory. Moreover, we explored the applications of this method in enantiomer ratio sensing by using the properties of the CD spectra. We found a strong linear dependence of the CD spectra on the enantiomer ratio. Our work provides a facile and efficient method to modulate the chirality of nanosystems, deepens our understanding of chiral plexcitons in nanosystems, and facilitates the development of chiral devices and chiral sensing.

## 1. Introduction

Chirality is widespread in nature, and the research on its origins and effects has sparked developments in areas such as biomedicine [1,2,3], stereochemistry [4], catalytic synthesis [5,6], and photonics [7,8]. Chiral substances and their mirror images (enantiomers) do not wholly coincide, and most of their basic properties are so similar that it is difficult to distinguish between them. However, confusing chiral enantiomers and being unable to distinguish between them can often lead to great disasters in the pharmaceutical, food, and life sciences [9,10]. Therefore, detecting, sensing, and recognizing chirality has been a hot research topic in the biomedical, chemical, and physical fields [11]. Generally, chiral molecules in nature, such as proteins and DNA, have a weak chiral optical response due to the mismatch with light waves caused by their smaller sizes [12]. In recent years, with the development of nano-synthesis technology, nano-processing, and fabrication technology, more and more chiral colloidal plasmonic nanoparticles (NPs) and nanostructures have been fabricated, which possess a strong chiral optical response, and have become the most promising new platforms for investigating chiral effects [13,14,15,16].

Colloidal plexcitonic materials (CPMs) are a class of nanosystems produced by the strong coupling between NPs and molecules [17], in which the components of NPs provide a powerful platform to strongly localize the electric field and exchange energy with excitons, resulting in the Rabi splitting and the formation of half-light–half-matter hybrid states when the rate of energy exchange exceeds the dissipation of the system [18,19,20,21]. In a Surface-Enhanced Circular Dichroism (SECD) nanosystem, NPs can be applied to enhance the CD spectra of chiral molecules because of their strong localized electric field. Notably, there is a systematic overlap between SECD systems and CPM systems [22,23,24]. However, previous studies of SECD systems did not generate chiral plexcitons due to the resonance mismatch between the chiral molecules and the achiral plasmonic nanoparticles, resulting in no novel CD spectral splitting being observed [7,25,26]. If chiral properties are introduced into a CPM system, for instance, an excitonic material with a chiroptical response or colloidal plasmonic nanoparticles with a chiral geometry, then due to the half-light–half-matter hybrid-state nature of the plexcitons, the entire CPM system will also have a chiroptical response, and be termed a chiral plexcitonic system [27,28,29]. Recently, the chiral plexcitonic system has become an attractive platform for studying strong light–matter interactions from a chiral spectroscopic perspective. The CD spectral splitting of CPMs was first observed experimentally by our group, which indicates the formation of a chiral plexcitonic system [28,29,30]. However, the chirality source of this chiral plexcitonic system is chiral molecules, which makes the chiroptical responses of the system very weak. Another strategy to generate chiral plexcitonic systems is to combine chiral NPs with achiral excitonic molecules, which significantly improves the chiroptical responses of the chiral plexcitonic system. Based on previous experimental and simulation works [30,31,32], the chiroptical responses are improved by approximately two orders of magnitude. For instance, helical nanorods [33,34] and nanorod dimers [35,36,37,38] were prepared for the fabrication of a chiral plexcitonic system. Despite these pioneering efforts, tuning the geometry of chiral NPs through synthesis strategies and assembly techniques remains a significant challenge.

Moreover, the complex geometries of chiral nanosystems do not allow us to easily reveal their electromagnetic response mechanisms through analytical methods. Therefore, there is great demand to develop methods based on extended models, and simulations to study the mechanism and unique advantages of chiral plexcitons for finely modulating the CD response, which will provide rules for controlling chiral light–matter interactions in complex chiral nanosystems.

In this paper, we present a chiral plexcitonic system that combines the concepts of chiral plasmonic nanoparticles and excitonic materials. Specifically, this chiral plexcitonic system consists of L-shaped nanorod dimers and achiral molecule excitons. Firstly, we calculated and analyzed the chiroptical response through finite element simulations and extended coupled-mode theory, and found that the formation of this chiral plexciton can significantly modulate the CD spectra of chiral plasmonic nanoparticles. Secondly, we found that the appearance of new hybridized peaks, double Rabi splitting, and bisignate anti-crossing behaviors in the CD spectra follow the properties of plexcitons in a strong-coupling regime. The bisignate feature of CD spectra allows the contributions of the various modes to be accurately differentiated. Finally, we explored the properties of the chiral plexcitonic system through the refractive index sensing and enantiomer ratio sensing of chiral samples, and found a strong linear dependence of the CD spectra on the enantiomer ratio ρ. Our work provides a facile and efficient method to modulate the chirality of nanosystems, deepens our understanding of chiral plexcitons in nanosystems, and will facilitate the development of chiral devices and chiral sensing.

## 2. Results and Discussion

### 2.1. Chiroptical Responses of the Chiral Plexcitonic System

Studying the chiroptical response of chiral plexcitons is a crucial step in designing familiar and easily accessible nanosystems composed of chiral plasmonic nanoparticles and J-aggregate molecules. As depicted in Figure 1a, we designed an L-shaped nanorod dimer to serve as a chiral plasmonic nanoparticle, which is widely applied in experiments as a model with 3D spatial chirality. In electromagnetic coupling analysis, this L-shaped nanorod dimer can be reduced using the Born–Kuhn model, introduced by Yin et al. [39] to describe the chiroptical response of this kind of chiral plasmonic nanoparticle. Specifically, two identical Au@Ag nanorods were assembled on opposite sides of a DNA origami sheet, and the long axes of the nanorods are perpendicular to each other. Then, to develop chiral plasmonic nanoparticles into CPMs, we coated the L-shaped nanorod dimer with J-aggregate molecules, which have low losses and a strong electric dipole moment in the visible region and are often used as exciton materials to form CPMs. Figure 1b,c show top and left views of the finalized chiral plexcitonic system, which facilitates our understanding of the spatial configuration of the system. A detailed description of the model and simulations is given in Appendix A.

The optical response of the L-shaped nanorod dimer is presented in Figure 2a, where we can clearly see the bimodal extinction spectra. That is caused by the near-field coupling of the two individual plasmonic nanorods, which results in two local surface plasmon resonance modes. The mode with high energy (642 nm) is the anti-bonding mode, and the mode with low energy (674 nm) is the bonding mode. The electric field distributions of these two modes have different characteristics (see Figure 3), which contribute to their different behavior in the spectra, such as resonance energy, resonance intensity, and loss. In particular, these two modes exhibit different resonance strengths when excited by right- and left-handed circularly polarized light (RCP and LCP), respectively; thus, a clear CD signal appears when the CD spectrum ($C_{Ext_{L}}-C_{Ext_{R}}$) in Figure 2b is obtained by calculating the difference in the extinction spectrum ($C_{Ext}=C_{abs}+C_{sca}$). These phenomena are also similar to some research results obtained using the Born–Kuhn model [39].

In general, it is rather challenging to adjust and reconfigure the chirality of chiral nanoparticles once they have been prepared or assembled, as in the case of the L-shaped nanorod dimer we discuss here, which can be assembled utilizing methods such as DNA origami [35,37] or atomic force microscopy (AFM)-based manipulation [16]. If one wants to change the chiroptical response through structural alterations, it inevitably involves steps such as recreating the DNA origami template, re-synthesizing nanorods with different aspect ratios and reassembling them, or AFM-based manipulation, which is usually prohibitively expensive and time-consuming. Through the strong chiral light–matter interaction between J-aggregate excitons and chiral plasmonic nanoparticles, we can significantly alter the chiroptical response of the system. On the experimental side, this method is easy to implement and avoids the complicated process of fabricating new samples, and on the theoretical side, the method can control the chiroptical responses of the system more accurately. As shown in Figure 2c,d, significant changes in CD spectra in terms of the appearance of new peaks and peak splitting can be observed, and these changes follow the properties of the chiral plexcitons. The phenomenon of strong coupling in these CD spectra also provides new spectroscopic research ideas for understanding strong light–matter coupling.

### 2.2. The Appearance of Double Rabi Splitting and Bisignate Anti-crossing Behaviors Follows the Extended Coupled-Mode Theory

As demonstrated in Figure 1a, chiral plexcitons can be generated by introducing chiral properties to the plasmon mode when the plasmon mode and the achiral exciton materials (J-aggregate excitons) interact strongly to form a CPMs. Analyzing the mechanism of this plexciton generation can also help us understand and reconfigure chiral nanoparticles’ chiroptical response. By analyzing and fitting the extinction spectrum of the L-shaped nanorod dimer, we can obtain the resonance frequencies and losses of the anti-bonding and bonding modes, as shown in Figure 3. Based on our understanding and theoretical calculations, the two modes independently interact with J-aggregates and produce plexcitons. Since there is a difference in these two modes under RCP and LCP light, the plexcitons will inherit this difference, thus naturally exhibiting a chiral response. Specifically, two anti-crossing behaviors with opposite signs emerged in the CD spectra. This phenomenon is rarely seen in the CD spectra of nanoparticles in previous studies. Based on the extended coupled-mode theory, we can describe the spectral splitting and anti-crossing behaviors with the following equations:(1)$$\left[\begin{array}{cccc} \omega_{aBM}-i\frac{\gamma_{aBM}}{2} & 0 & g_{1} & 0\\ 0 & \omega_{BM}-i\frac{\gamma_{BM}}{2} & 0 & g_{2}\\ g_{1} & 0 & \omega_{ex}-i\frac{\gamma_{ex}}{2} & 0\\ 0 & g_{2} & 0 & \omega_{ex}-i\frac{\gamma_{ex}}{2} \end{array}\right]\left[\begin{array}{c} \alpha_{1}\\ \alpha_{2}\\ \alpha_{3}\\ \alpha_{4} \end{array}\right]=\omega \left[\begin{array}{c} \alpha_{1}\\ \alpha_{2}\\ \alpha_{3}\\ \alpha_{4} \end{array}\right]$$
where $\omega_{aBM}$, $\omega_{BM}$, and $\omega_{ex}$ are the anti-bonding mode, bonding mode, and J-aggregate exciton energies, and $\gamma_{aBM}$, $\gamma_{BM}$, and $\gamma_{ex}$ are their dissipation rates. The coupling strengths between the anti-bonding mode (bonding mode) and excitons are represented by $g_{1}$ ($g_{2}$). $\alpha_{i}$ represents the Hopfield coefficients, and their square represents the proportion of the plasmon mode and exciton energy in the plexcitons. The plexciton energy ω can be calculated from the above equation:
(2a)$$\begin{array}{cc} \omega_{aBM\pm } & =\frac{\omega_{aBM}+\omega_{ex}}{2}-i\frac{\gamma_{aBM}+\gamma_{ex}}{4}\pm \sqrt{g_{1}^{2}+\frac{\delta_{1}^{2}}{4}} \end{array}$$
(2b)$$\begin{array}{cc} \omega_{BM\pm } & =\frac{\omega_{BM}+\omega_{ex}}{2}-i\frac{\gamma_{BM}+\gamma_{ex}}{4}\pm \sqrt{g_{2}^{2}+\frac{\delta_{2}^{2}}{4}} \end{array}$$

Table 1 shows the values of the parameters in Equations (1) and (2), where the values of $\omega_{aBM}$, $\omega_{BM}$, $\gamma_{aBM}$, $\gamma_{BM}$, and $\gamma_{ex}$ are fitted by the simulated extinction spectra in Figure 3a. We substitute the parameters $\omega_{aBM}$, $\omega_{BM}$, $\omega_{ex}$, $\gamma_{aBM}$, $\gamma_{BM}$, and $\gamma_{ex}$ into Equation (2a,b), so that $g_{i}$ (i = 1, 2) is the only variable parameter, and then calculate the energy distribution of the hybridized states to fit with the simulation results. The value of $g_{i}$ can be obtained when the theoretical and simulation results are perfectly matched. The energy detuning between the plasmon mode and excitons is represented as $\delta_{1}=\left(\omega_{aBM}-i\gamma_{aBM}/2\right)-\left(\omega_{ex}-i\gamma_{ex}/2\right)$, $\delta_{2}=\left(\omega_{BM}-i\gamma_{BM}/2\right)-\left(\omega_{ex}-i\gamma_{ex}/2\right)$. The Rabi splitting $\Omega =\omega_{+}-\omega_{-}$. In the zero-detuning case ($\omega_{aBM}=\omega_{ex}$ or $\omega_{BM}=\omega_{ex}$), the Rabi splitting can be solved as $\Omega_{aBM}=\sqrt{4g_{1}^{2}-\frac{{\left(\gamma_{aBM}-\gamma_{ex}\right)}^{2}}{4}}$ or $\Omega_{BM}=\sqrt{4g_{2}^{2}-\frac{{\left(\gamma_{BM}-\gamma_{ex}\right)}^{2}}{4}}$. The CD spectra do not show intermediate states due to the uncoupling between the bonding and antibonding modes. The CD spectra behave more as a combination of two strong coupled systems, unlike the common three-mode polariton coupling, where intermediate coupling states occur [40].

As shown in Figure 3d, we calculated the energy distribution of the plexcitonic system according to Equation (2), and it can be observed that the bonding mode and the anti-bonding mode are strongly coupled with the excitons. Furthermore, the anti-crossing behavior and the Rabi splitting $\Omega_{BM}=98.5$ meV, $\Omega_{aBM}=106.1$ meV can be obtained. In our system, the dissipation rates of the plasmon mode and excitons are $\gamma_{aBM}=121$ meV ($\gamma_{BM}=91$ meV) and $\gamma_{ex}=50$ meV. The criterion of a strong-coupling regime, $\Omega_{aBM}>\left(\gamma_{aBM}+\gamma_{ex}\right)/2=85.5$ meV and $\Omega_{BM}>\left(\gamma_{BM}+\gamma_{ex}\right)/2=70.5$ meV, is satisfied. The anti-crossing behavior of the spectra and the strong-coupling criterion both suggest that the chiral plexcitonic system attained a strong-coupling regime. In Figure 3b,c, the energy distributions of the chiral plexcitonic system on the extinction and CD spectra are also illustrated by finite element simulation calculations, and it can be seen that two obvious instances of Rabi splitting occur in the yellow positive and blue-violet negative regions of the CD spectra, respectively.

This agrees well with the theoretical calculation in Figure 3d. However, only the Rabi splitting of the bonding mode is apparent in the extinction spectra in Figure 3b. A similar phenomenon was observed in the research of Zhu et al. [38]. Here, we can presume that this is because BM has a stronger local field (see Appendix A
Figure A1), so the strong-coupling criterion is easier to satisfy. The anti-crossing behavior is more straightforward to observe in the spectra. In contrast, the aBM and BM resonances are relatively close to each other, as shown in Figure 3a, which leads to another anti-crossing phenomenon of aBM hidden in the extinction spectra. This is why only one instance of Rabi splitting is apparent in the extinction spectra in Figure 3b. It is also implied that the overlap of some modes can seriously hamper the observation and analysis of strong-coupling phenomena in the extinction spectra. In contrast, the CD spectra can distinguish the effects of these modes. In a way, it is a handy and practical spectroscopic measurement to observe strong-coupling phenomena through CD spectra.

Furthermore, as shown in Figure 4a–d, we calculated the Hopfield coefficients using Equation (Equation 1), which reflects the energy proportion of each component in the chiral plexcitons. There is no chiral response of excitons in the system, and all the chirality originates from the plasmon mode. Thus, the higher the percentage of plasmon components in the chiral plexcitons, the stronger the chiral response. For example, in Figure 4a–d, the plasmon proportion of aBM+ and BM+ shows a decreasing trend, and the plasmon proportion of aBM− and BM− shows an increasing trend. From Figure 3c, we can observe that the CD intensity variation trend of each branch of aBM+, aBM−, BM+, and BM− also shows a similar principle regarding the proportion of plasmon. The excitons in our system are achiral, and the strong-coupling interactions have the effect of transferring the chirality; specifically, the effect is to transfer the CD response of the chiral particles to the position of the hybrid peaks formed by the strong coupling. From this perspective, the chiral optical field confers the CD response to the achiral exciton, so the CD of the achiral exciton is connected to the chiral optical field. This indicates that the emergence of new peaks in the chiral plexcitonic system is governed by strong-coupling effects, and the intensity of CD is also influenced by the plasmon proportion of the chiral plexcitonic system.

### 2.3. Applications in Enantiomer Ratio Sensing by Using the Properties of the CD Spectra of the Chiral Plexcitonic System

Some nanostructures inherently produce chiral fields. For example, the plasmonic spiral shapes studied by Zhou et al. [41] and Zaman et al. [42] not only generate vortex light, but also can be used for sensing applications. Since a chiral plexcitonic system has the properties of the plasmons and an inherent chiral field, it is natural to wonder whether there are some properties of this system that are useful for refractive index sensing and chiral molecular component sensing as well. As shown in Figure 5, we calculated the optical chirality (OC) distribution of the nanorod dimer system (details of the methods of calculation are given in the Section A.2). Indeed, optical chirality (OC) is an important physical quantity that measures the chirality of local electric and magnetic fields, which can characterize the enhancement of circular dichroism in chiral molecule detection [43]. The color distribution in Figure 5 represents the enhancement factor of the optical chirality, in which we can achieve up to a 250-fold enhancement (compared to the OC of CPL). The strongest regions of the enhancement factor are distributed near the ends of the nanorods, and these sites also capture chiral molecules easily in the experiment. This means that our system is well suited to recognizing and detecting chiral molecules. Basically, the chiral sample, such as chiral molecular aggregates and chiral quantum dots, can be called the Pasteur medium (a non-magnetic bi-isotropic medium) and modeled as a homogeneous layer, whose constitutive relationships are expressed [44] as $D=\varepsilon \varepsilon_{0}E-i\kappa \sqrt{\mu_{0}\varepsilon_{0}}\;H$, $B=\mu_{0}H+i\kappa \sqrt{\mu_{0}\varepsilon_{0}}\;E$. Here, $\varepsilon_{0}$, $\mu_{0}$, and κ are permittivity, permeability, and Pasteur parameters, respectively. Specifically, ε is governed by the concentration of the chiral sample, while κ is governed by the enantiomer ratio ρ. The relationship can be written as $\varepsilon =\varepsilon_{b}+\alpha f\left(\omega \right)$, κ=(2ρ−1)βh(ω), where αf(ω) and βh(ω) are Lorentzian parameters related to the electronic transitions and are specific to the molecular species. The details of implementing these expressions of the chiral samples in the numerical simulations can be found in Appendix A. In Figure 6a,b, we chose the chiral plexcitonic system discussed above and replaced its background environment with a chiral sample. Due to the high proportion of plasmon components in chiral plexcitons during red- and blue-detuning, it is very sensitive to refractive index variations. All the peaks in Figure 6a exhibit redshifts with increasing refractive index, and the intensities of the two peaks with positive signals exhibit decreasing and increasing trends, respectively. The negative peak has the strongest intensity and exhibits a refractive index sensitivity of S=301.6 nm/RIU. A similar result is observed in Figure 6b, where the peak with positive values has a refractive index sensitivity of S=320.4 nm/RIU. For a chiral sample with an enantiomer ratio ρ, we calculated the ΔCD, the difference between the CD of the chiral sample and the CD of the racemic sample (ρ=50%). In Figure 6c, during a ρ increase (an increase in the enantiomer ratio of the sample), the peak at 705 nm exhibits a tendency to decrease to zero, and then the signal reverses its sign and continues to increase in absolute value. Moreover, as shown in Figure 6d, the peak value of the ΔCD spectrum follows a linear relationship for ρ. Thus, it is convenient to evaluate the enantiomer ratio of a chiral sample through this linear relationship.

## 3. Conclusions

In summary, we investigated the mechanism of reconfigurable chirality and chiral sensing based on chiral plexcitons in a strong-coupling regime. We designed a chiral plexcitonic system consisting of L-shaped nanorod dimers and achiral molecule excitons. We calculated and analyzed the chiroptical responses through finite element simulations and extended coupled-mode theory, and found that the formation of these chiral plexcitons can significantly modulate the spectra of chiral plasmonic nanoparticles, which is much more efficient and simple than the method of directly changing the particle morphology. The appearance of new hybridized peaks and spectral splitting follows the properties of plexcitons in the strong-coupling regime, which provides a rule for modulating the chiroptical responses of the chiral plexcitonic system. We demonstrated that double Rabi splitting and bisignate anti-crossing behaviors can be observed in circular dichroism (CD) spectra. Meanwhile, the bisignate feature of CD spectra allows the contributions of various modes to be accurately differentiated. Additionally, we explored the properties of the chiral plexcitonic system in refractive index sensing and enantiomer ratio sensing in chiral samples. We found a strong linear dependence of the CD spectra on the enantiomer ratio. In the future, our research will experimentally explore the rich chiral optical properties of chiral plexcitonic systems from a multi-dimensional and complex perspective. Our research provides a facile and efficient method to modulate the chirality of nanosystems, deepen our understanding of chiral plexcitons in nanosystems, and facilitate the development of chiral devices and chiral sensing.

## Figures and Tables

**Figure 1 nanomaterials-14-00705-f001:**
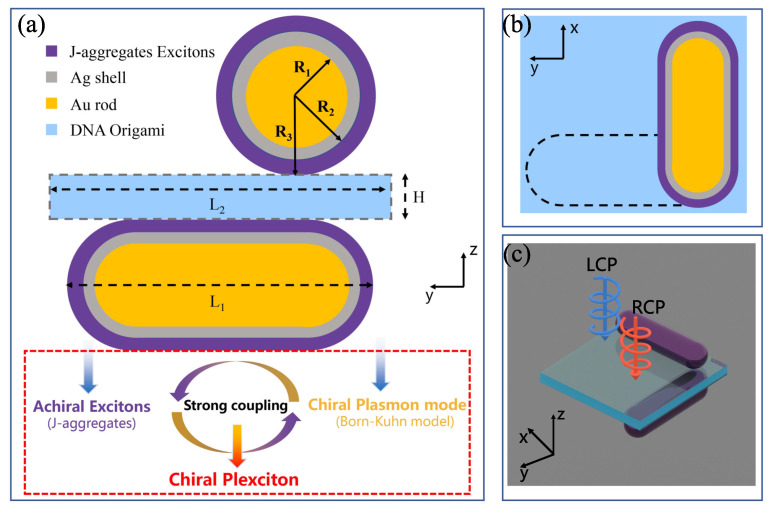
Schematic representation of the design of the chiral plexcitonic system. (**a**) Front view of the chiral plexcitonic system (yz-plane), where the yellow parts represent gold nanorods, the gray parts represent silver shells, and the blue part represents the DNA origami sheet. Two identical Au@Ag nanorods were assembled on the opposite surfaces of the DNA origami sheet, and then, the J-aggregate molecules (red parts) were adsorbed on the Au@Ag nanorods’ surfaces. (**b**,**c**) represent the top (xy-plane) and 3D views, respectively. The L-shaped nanorod dimers form a typical chiral plasmonic nanostructure (Born–Kuhn model) and strong coupling with the achiral excitons (J-aggregates). This complex process forms CPMs with a chiroptical response, which we denominate as chiral plexcitons. Here, $R_{1}=7.5$ nm, $R_{2}=13$ nm, $R_{3}=16$ nm, $L_{1}=60$ nm, $L_{2}=70$ nm, and H=12 nm.

**Figure 2 nanomaterials-14-00705-f002:**
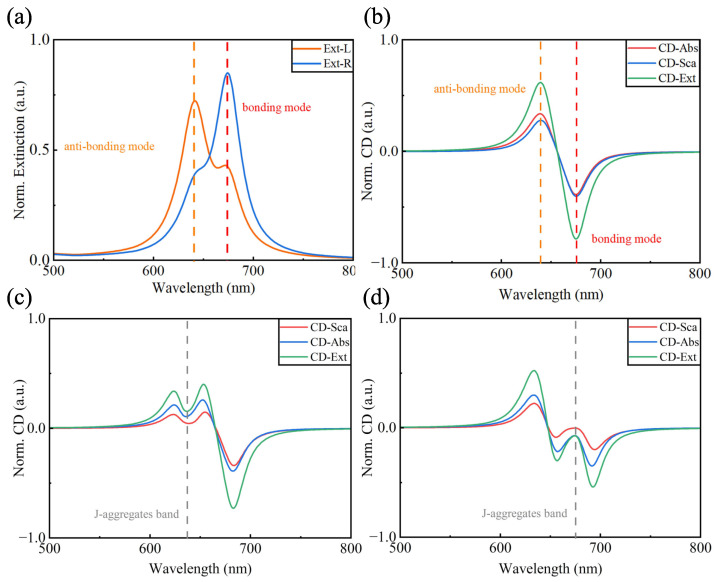
Extinction spectra and CD spectra of the chiral plexcitonic system. (**a**) Extinction spectra of L-shaped nanorod dimer for circularly polarized light incident along the z-axis, with orange and blue ranges representing the incident cases of RCP and LCP, respectively, where the peak of the orange dashed line is called the anti-bonding mode and the peak of the red dashed line is called the bonding mode. (**b**) shows the CD spectra of the L-shaped nanorod dimer, where the green solid line is the difference in the extinction spectrum, the red solid line is the difference in the absorption spectrum, and the blue solid line is the difference in the scattering spectrum under RCP and LCP. (**c**,**d**) show the CD spectra of the chiral plexcitonic system (L-shaped nanorod dimers strongly coupled with J-aggregate molecules). The resonance energies of the J-aggregate molecules are in the vicinity of the anti-bonding and bonding modes, respectively.

**Figure 3 nanomaterials-14-00705-f003:**
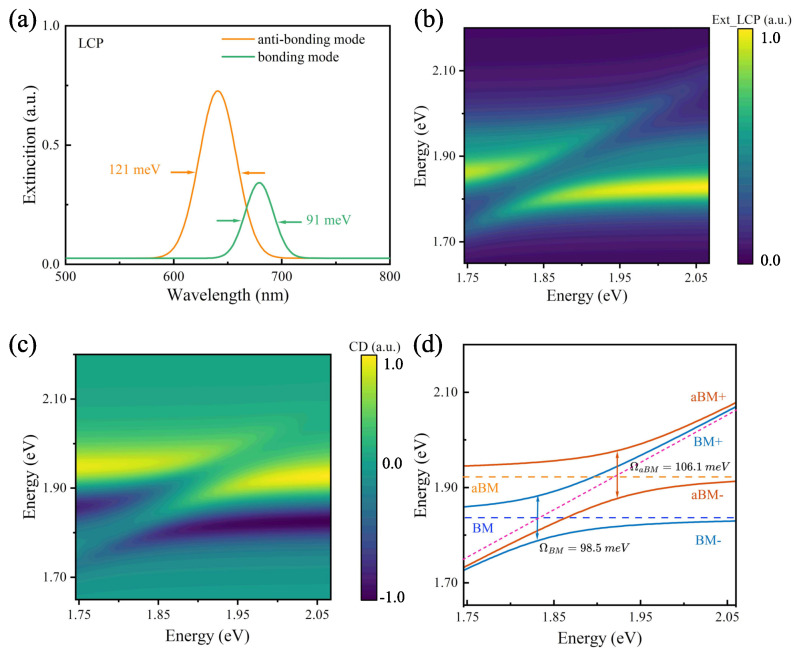
Extinction spectra and CD spectra of the chiral plexcitonic system. (**a**) Extinction spectra of anti-bonding and bonding modes, calculated by Lorentz fittings. The solid orange line represents anti-bonding mode with a dissipation of $\gamma_{aBM}=121$ meV; the solid green line represents bonding mode with a dissipation of $\gamma_{BM}=91$ meV. (**b**,**c**) depict the evolution of the plexciton state energies in the extinction and CD spectra when the J-aggregate excitons’ energy is detuned with the energy of the plasmon modes. The horizontal axis represents the exciton energy and the vertical axis represents the plasmon energy. (**d**) shows an energy dispersion diagram of the plexciton state calculated via Equation (2). The orange and blue dashed lines indicate the energies of aBM and BM, respectively, while the orange and blue solid lines indicate the energies of the plexcitons formed by aBM and BM, respectively.

**Figure 4 nanomaterials-14-00705-f004:**
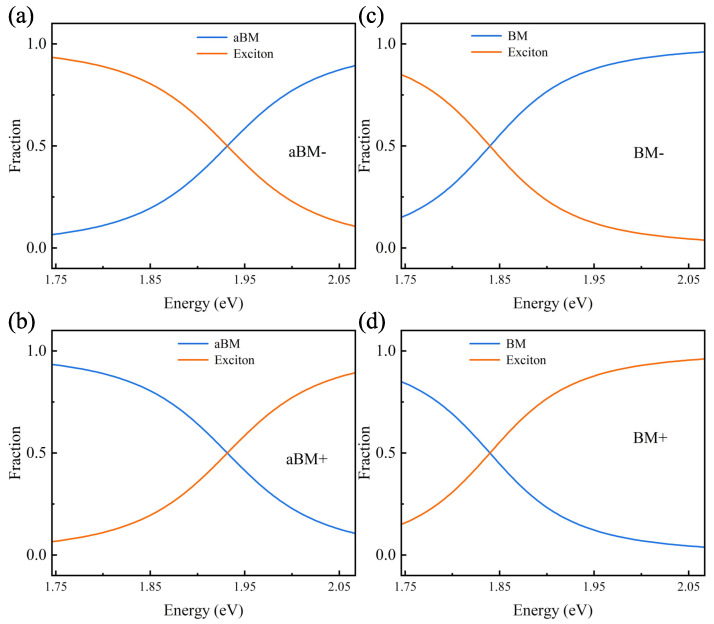
Hybrid fraction (Hopfield coefficients) of the chiral plexcitonic system calculated from Equation (Equation 1). (**a**,**b**) depict the proportion of excitons and plasmons in the aBM− and aBM+ energy branches; (**c**,**d**) depict the proportion of exciton and plasmons in the BM− and BM+ energy branches. The orange and blue solid lines indicate the hybrid energies of the excitons and plasmons, respectively.

**Figure 5 nanomaterials-14-00705-f005:**
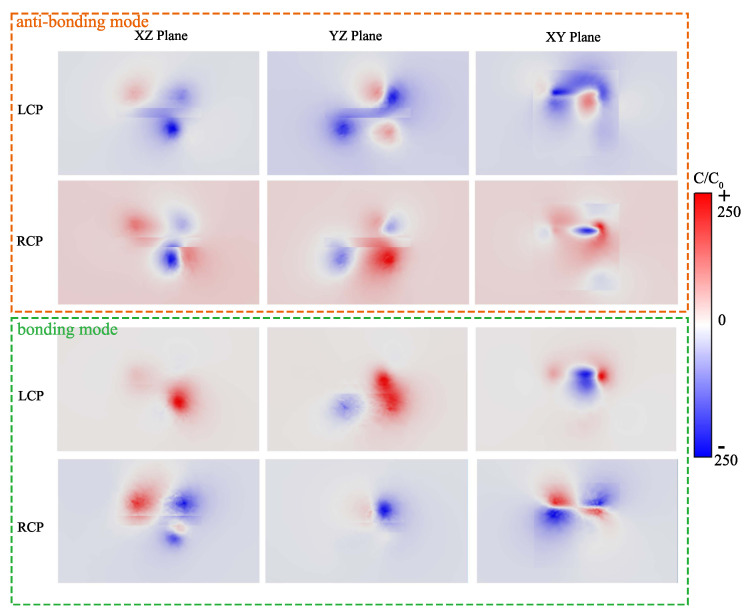
The optical chirality (OC) distribution within the nanorod dimer system. The orange rectangular box shows the OC distributions in the XZ−plane, YZ−plane, and XY−plane for anti-bonding mode under LCP and RCP excitations, respectively. The green rectangular box shows the OC distributions in the XY−plane, YZ−plane, and XY−plane for bonding mode under LCP and RCP excitations, respectively. $C/C_{0}$ is the OC density enhancement factor, where $C_{0}$ is the OC density of CPL alone.

**Figure 6 nanomaterials-14-00705-f006:**
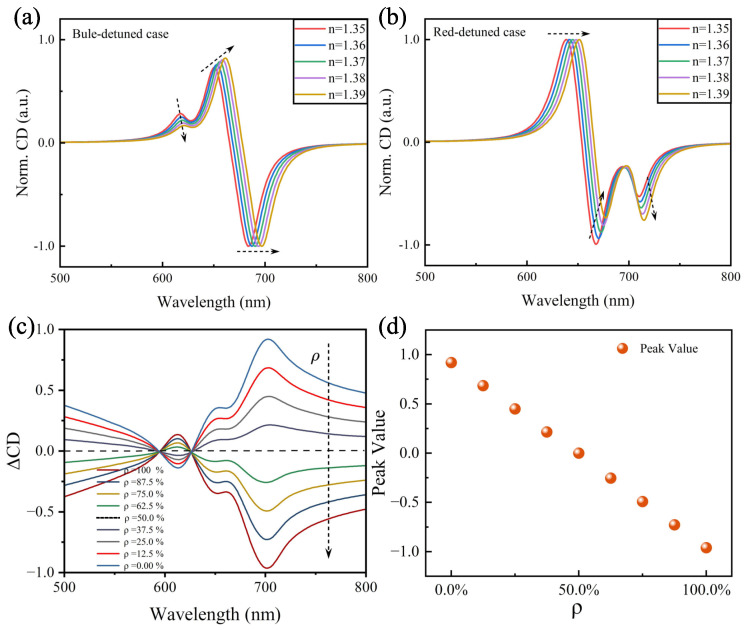
Influence of refractive index *n* and enantiomer ratio ρ of chiral sample on the CD spectrum of the chiral plexcitonic system. (**a**,**b**) show the CD spectra of the chiral plexcitonic system in blue- and red-detuned cases, where *n* increases from 1.35 to 1.39. The sensitivity values are found to be S=301.6 nm/RIU and S=320.4 nm/RIU. (**c**) shows the ΔCD spectra of the chiral plexcitonic system with an increasing enantiomer ratio ρ. (**d**) Peak value of ΔCD spectrum as a function of ρ, showing a linear relationship.

**Table 1 nanomaterials-14-00705-t001:** Parameters of Equations (1) and (2).

$\omega_{aBM}$ [eV]	$\omega_{BM}$ [eV]	$\gamma_{aBM}$ [eV]	$\gamma_{BM}$ [eV]	$\gamma_{ex}$ [eV]	$g_{1}$ [eV]	$g_{2}$ [eV]
1.9315	1.8398	0.121	0.091	0.050	0.04275	0.03525

## Data Availability

All data are available from the authors upon reasonable request.

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
