# Peer review of "The Mechanism of Manipulating Chirality and Chiral Sensing Based on Chiral Plexcitons in a Strong-Coupling Regime"

_nanomaterials, 2024, doi:10.3390/nano14080705_

Round 1
Reviewer 1 Report (Previous Reviewer 1)
Comments and Suggestions for Authors
The Authors have improved the manuscript significantly. However, a few questions/problems remain that I would like to see addressed in the manuscript before publication. I would suggest that the manuscript be published in nanomaterials after addressing the following points.
[1] In addition to the OC distributions, its spectra (wavelength dependence) also include important information. Comparing near-field OC spectra and far-field CD spectra deepens the discussion of the chiroptical response of nanostructures.
[2] In the response of comment 1-3, the authors explain that they study the chiral plasmon/achiral exciton system because the chiroptical response of the achiral plasmon/chiral exciton system is relatively weak. However, they can discuss the differences between these systems more quantitatively.
Author Response
Please see the attachment.

Reviewer 2 Report (Previous Reviewer 3)
Comments and Suggestions for Authors
The paper has been improved. The authors have addressed most of my queries. The paper can be considered for publication.
My only recommendation is that the authors include a 3D diagram of the nanorod dimers in the main text (similar to the top figure of R2 in the review response, but of perhaps of better quality like figure R1 c). This will make the 3D chirality of the L-shape dimers more clear for the readers.
Comments on the Quality of English Language
Enlgish is fine.
Author Response
Please see the attachment.

This manuscript is a resubmission of an earlier submission. The following is a list of the peer review reports and author responses from that submission.
Round 1
Reviewer 1 Report
Comments and Suggestions for Authors
The authors simulated a coupling regime between chiral plasmon mode and J-aggregates. The simulation suggested that the plexcitons of the chiral plasmon and excitons showed large chiroptical responses. The novelty of this study is not clear because a similar system has been reported with experimental results. Thus, the authors should show a significant improvement using numerical simulations. Additionally, I have several concerns, as shown below.
I think that significant work is needed to bring this study to the level required for publication in nanomaterials.
[Specific comments]
[1] The authors focus on the simulation of far-field optical spectra under a strong coupling regime. These are easily performed by experiment. However, the chiral interactions in near-field are also important and worth being confirmed by electromagnetic simulation. Electric field, optical chirality and these spectra should be investigated.
[2] In Figure 4c, the CD of plexcitons depends on the chirality of the excitons, although the origin of chiroptical responses is explained as the chiral geometry of plasmonic particles. It seems that the CD response of excitons is simply enhanced by the near-field enhancement independently of the chiral optical field.
[3] Essential control data are lacking. Although data of chiral plasmon/achiral excitons and chiral plasmon/chiral excitons are shown, more data of achiral plasmon/achiral excitons and achiral plasmon/chiral excitons should be added.
Reviewer 2 Report
Comments and Suggestions for Authors
In this manuscript, the authors present their theoretical research on chiral plexitonic materials. In the current study, Finite Element Method was employed in investigating light-matter interactions between circularly polarized light and chiral assembly of gold/silver core-shell nanorods assembled on the opposite surfaces of DNA origami sheet coupled with J-aggregates.
The research is new and important, as chiral sensing and manipulation has been a growing subject for years and has attracted significant attention of the scientific community, as the authors point out.
The reviewer finds this manuscript very well organized and written, the conclusions are supported with the results derived from FEM calculations. The overall quality of this article is high, thus the reviewer recommends the editor to accept the article with a few minor corrections.
Figure 3 caption has to be rewritten, as it has a lot of spelling errors. “cacaulted” and “dispission” are just two examples of such mistakes.
The parameters given at the end of Appendix A are “reasonably assumed to be” some values. The authors should explain why such values were chosen and if they have been previously used in similar calculations, they should include proper reference.
Thus, the reviewer recommends the editor to accept the manuscript after small modifications.
Reviewer 3 Report
Comments and Suggestions for Authors
The paper presents a numerical study on L-shaped nanorod dimers that form a chiral plexcitons system. Along with numerical finite element analysis results, a coupled-mode-theory analysis is presented. The bonding and anti-bonding modes along with Rabi splitting is discussed. Numerical analysis on a possible application in enantiomer ratio sensing is presented. While the sensing results are clear and well stated, the rest of the paper is not as clearly described. The organization of the paper does not flow smoothly. Firstly, the introduction section lacks details about the novelty and contribution of the work. A clearer discussion is needed. In the literature review, papers related to numerical study should be the main focus. As the work is numerical, it should not be compared (or done so only briefly) with experimental work. Next, each analysis section should clearly state why that analysis is needed/important. That will significantly improve the readability of the paper. The coupled-mode theory should be discussed in more detail and it should be mentioned how each of the parameters (such as the coupling strengths) can be calculated. The parameters needed to recreate Fig. 3 should be provided. The numerical results alone might not be sufficient for the paper to be publishable in nanomaterials. Hence, the coupled-mode-theory part should be strengthened. I recommend major revisions the paper the can be considered for publication. My detailed comments are listed below:
1. The novelty of the paper is not clearly stated. Chiral plasmonic nanoparticles, J-aggregates, tunable plexcitons, L-shaped nanorod dimers etc. have all been previously reported in the literature (as is cited in the paper). A clear concise paragraph is needed in the introduction section to state the novelty and contribution of the paper. Perhaps the current literature review part can be somewhat shortened for brevity.
2. In Fig. 2(a), the extinction spectra are drawn using filled semi-transparent curves. This is unnecessary and actually reduces the clarity of the data. Please use standard solid unfilled curves (as in Fig. 2(b), (c) and (d)).
3. When reviewing reference [13-16], the paper states that chiral nanoparticles are complex and expensive to fabricate. However, it is not clear how the proposed approach is an improvement over this. Please clarify.
4. The paper is purely numerical with no experimental results. Hence, it is strange when the paper states that the L-shaped nanorod dimers are assembled using DNA origami. The DNA origami plays no role in the numerical model and would only be relevant for an experimental work. It might be acceptable to state that “such L-shaped dimers can be assembled using DNA origami” which would convey the sense that the current paper did not carry out the process. The DNA origami phrases used through out the current manuscript is misleading and should be thoroughly revised.
5. L-shaped structures are only chiral in 2D. In a colloidal system, an L-shaped can rotate in the out-of-plane direction to create its mirror image. How would one handle such a situation in an experimental study? Please comment.
6. How are the coupling strengths g1 and g2 in Eq. (1) calculated?
7. Assuming Fig. 3(a),(b),(c) and (d) are generated using Eq. 1, 2a and 2b, please provide the numerical values of all parameters so that others can reproduce the plots. Also, please mention how these parameter values were obtained/calculated so that the readers can extend the analysis to other systems.
8. At what wavelength is Fig. 5d generated? Do the peaks in Fig. 5c occur at the same wavelengths for all cases? In practice, what kind of light source would be used to excite such a broadband response?
9. Some plasmonic structures/geometries inherently produce chiral fields. For example, spiral shapes. Such structures are easier to fabricate and can be used for sensing applications. Please briefly discuss this in the manuscript. A few papers on such plasmonic resonators should be cited:
- https://doi.org/10.1007/s11467-020-1032-y
- https://doi.org/10.1103/PhysRevA.100.013857
Comments on the Quality of English Language
English is fine.